# The anti-arthritis effect of sulforaphane, an activator of Nrf2, is associated with inhibition of both B cell differentiation and the production of inflammatory cytokines

**Su-Jin Moon[1][☉], Jooyeon Jhun[2][☉], Jaeyoon Ryu[2], Ji ye Kwon[2], Se-Young Kim[2], KyoungAh Jung[3], Mi-La Cho [2,3,4][‡]*, Jun-Ki Min[5][‡]***

1 Division of Rheumatology, Department of Internal Medicine, Uijeongbu St. Mary's Hospital, College of Medicine, The Catholic University of Korea, Uijeongbu, South Korea, 2 The Rheumatism Research Center, Catholic Research Institute of Medical Science, The Catholic University of Korea, Seoul, South Korea, 3 Impact Biotech, Seoul, South Korea, 4 Laboratory of Immune Network, Conversant Research Consortium in Immunologic Disease, College of Medicine, The Catholic University of Korea, Seoul, South Korea, 5 Department of Internal Medicine, and the Clinical Medicine Research Institute of Bucheon St. Mary's Hospital, Bucheon-si, South Korea

☉ These authors contributed equally to this work.
‡ MLC and JKM also contributed equally to this work.
* rmin6403@hanmail.net (JKM); iammila@catholic.ac.kr (MLC)

**Data Availability Statement:** All relevant data are within the manuscript.

## Abstract

Nuclear factor (erythroid-derived 2)-like 2 (Nrf2) is an important transcription factor that plays a pivotal role in cellular defense against oxidative injury. Nrf2 signaling is involved in attenuating autoimmune disorders such as rheumatoid arthritis (RA). B cells play several roles in the pathogenesis of RA, such as in autoantibody production, antigen presentation, and T-cell activation. We investigated the anti-arthritic mechanisms of sulforaphane, an activator of Nrf2, in terms of its effect on B cells. To investigate the effect of sulforaphane on collagen-induced arthritis (CIA), sulforaphane was administered intraperitoneally after CIA induction. Hematoxylin and eosin-stained sections were scored for inflammation, pannus invasion, and bone and cartilage damage. We assessed the expression levels of inflammation-related factors by real-time PCR and the levels of various IgG subclasses by enzyme-linked immunosorbent assay. Sulforaphane treatment reduced the arthritis score and the severity of histologic inflammation in CIA mice. The joints from sulforaphane-treated CIA mice showed decreased expression of interleukin (IL)-6, IL-17, tumor necrosis factor (TNF)-α, receptor activator of NF-κB ligand, and tartrate-resistant acid phosphatase. Sulforaphane-treated mice showed lower circulating levels of type-II-collagen-specific IgG, IgG1, and IgG2a. *In vitro*, sulforaphane treatment significantly reduced the differentiation of lipopolysaccharide-stimulated murine splenocytes into plasma B cells and germinal-center B cells. Finally, sulforaphane significantly inhibited the production of IL-6, TNF-α, and IL-17 by human peripheral blood mononuclear cells stimulated with an anti-CD3 monoclonal antibody in a dose-dependent manner. Inhibition of differentiation into plasma B and Germinal Center B cells may be the mechanism underlying the anti-arthritic effect of sulforaphane.

**Funding:** This study was supported by a grant of the Korean Health Technology R&D Project, Ministry for Health & Welfare, Republic of Korea (HI14C1851). This work was supported by the National Research Foundation of Korea(NRF) grant funded by the Korea government(MSIT) (No. NRF-2018R1C1B6005854). This work was supported by the National Research Foundation of Korea (NRF) grant funded by the Korea government (MSIT) (No. NRF-2018R1A2B6007648).

**Competing interests:** The authors declare that there is no conflict of interest regarding the publication of this article.

**Abbreviations:** RA, rheumatoid arthritis; CIA, collagen induced arthritis; Nrf2, Nuclear factor erythroid 2-related factor 2; TNF-α, tumor necrosis factor-α; IL, interleukin; CCP, citrullinated peptide; RF, rheumatoid factor.

# Introduction

Rheumatoid arthritis (RA) is a chronic inflammatory disease that is characterized by infiltration of immune cells (such as T and B cells) into the hyperplastic synovium, which manifests as pannus formation. The disease eventually leads to progressive joint destruction and reduces quality of life. The pathogenesis of RA involves various immune cells and inflammatory cytokines. B cells are phylogenetically the most recent evolutionary development in the immune system. The importance of B cells and plasma cells in the development and progression of RA was suggested [1] by detection of autoantibodies such as anti-cyclic citrullinated peptide (anti-CCP) and anti-rheumatoid factor (anti-RF) in patients with RA [2, 3]. In addition to its diagnostic utility, the presence of autoantibodies suggests a poor prognosis of RA, such as rapid joint destruction [4]. The fact that autoantibodies are detected in the majority of RA patients up to 14 years before disease onset implies that B cells are essential for its pathogenesis [5]. Furthermore, synovial B cells and plasma cells of RA patients show evidence of antigen-driven maturation and autoantibody production [6]. As autoantibody production leads to immune complex formation and cytokine release [7], plasma cell hyperreactivity is viewed as a key component of the development and perpetuation of RA [8, 9]. Therefore, B-cell depletion is used to treat RA [10, 11].

Nuclear factor (erythroid-derived 2)-like 2 (Nrf2) is a master regulator of cellular protective processes and modulates the expression of several antioxidant and anti-inflammatory genes. Nrf2 activation may regulate the production of factors implicated in the pathophysiology of rheumatic diseases, including RA [12], lupus nephritis [13], and osteoarthritis [14]. There has been little research on the effect of Nrf2 activity on B cells. One study examined the effect of Nrf2 activity on B cells in an animal model of chronic lung inflammation [15].

In this study, plasma-cell infiltration was significantly greater in Nrf2-/- compared with wild-type mice [15]. Here, we determined whether the anti-arthritic effect of sulforaphane in an animal model of RA is mediated by modulation of B-cell differentiation and activity.

# Material and methods

## Mice

Seven-week-old male DBA/1J mice (Orient Bio, Gyeonggi-do, Korea) were maintained under specific pathogen-free conditions and fed standard laboratory mouse chow (Ralston Purina, St. Louis, MO, USA) and water *ad libitum*. The animals were housed five per cage in a room maintained under controlled temperature (21–22°C) and lighting (12 h light/dark cycle) conditions. All experimental procedures were approved by the Institutional Animal Care and Use Committee at the School of Medicine and the Animal Research Ethics Committee of the Catholic University of Korea and were conducted in accordance with the Laboratory Animals Welfare Act according to the Guide for the Care and Use of Laboratory Animals. All experimental procedures were evaluated and conducted in accordance with the protocols approved by the Animal Research Ethics Committee at the Catholic University of Korea (Permit Number: CUMC 2016-0086-01). All procedures performed in this study followed the ethical guidelines for animal use.

## Clinical scoring of CIA

CIA was induced in DBA1/J mice (n = 5). The experiment was performed three times. Mice were observed twice weekly for the onset, duration, and severity of joint inflammation for 7 weeks after the primary immunization. The mice were considered to have arthritis when significant changes in redness and/or swelling were noted in the digits or in other parts of the

paws. Knee-joint inflammation was scored visually after dissection on a scale from 0 to 4 (0, normal; 1, mild swelling; 2, moderately severe arthritis involving toes and ankle; 3, severe arthritis involving an entire paw; and 4, severe arthritis). Scoring was performed by two independent observers.

## Preparation of type II collagen Ag and immunization

CIA was induced in DBA1/J mice (n = 5). The experiment was performed three times. Male DBA/1J mice were immunized intradermally at the base of the tail with 100 μg of bovine type II collagen (CII) emulsified in complete Freund's adjuvant (CFA; Arthrogen-CIA, Redmond, WA, USA) (1:1, w/v). Two weeks later, the mice were boosted by intradermal injection with 100 μg of bovine CII in incomplete Freund's adjuvant (IFA; DIFCO, Detroit, MI, USA) (1:1, v/v). Three weeks after the primary immunization, collagen-induced arthritis (CIA) mice were injected intraperitoneally with 1.5 mM sulforaphane (200 μL of sulforaphane at 12.8 mg/mL/kg) every other day for 7 weeks. Control mice received vehicle (phosphate buffered saline) alone.

## Histological assessment of arthritis

The hematoxylin and eosin-stained sections were scored for inflammation and cartilage damage. Inflammation was scored as follows: score 0, no inflammation; score 1, slight thickening of the lining layer or some infiltrating cells in the sublining layer; score 2, slight thickening of the lining layer plus some infiltrating cells in the sublining layer; score 3, thickening of the lining layer, influx of cells in the sublining layer, and presence of cells in the synovial space; and score 4, synovium highly infiltrated with many inflammatory cells. Scoring of cartilage erosion was performed as follows: score 0, no destruction; score 1, minimal erosion limited to single spots; score 2, slight-to-moderate erosion in a limited area; score 3, extended erosions; and score 4, general destruction. Neutrophils were enumerated in three adjacent sections.

## Immunohistochemistry

Immunohistochemistry was performed using the Vectastatin ABCkit (Vector Laboratories, Burlingame, CA, USA). Joint tissue of sulforaphane- and vehicle-treated mice was incubated with primary antibodies to interleukin (IL)-6, IL-17, tumor necrosis factor (TNF)-α, receptor activator of NF-κB ligand (RANKL), and tartrate-resistant acid phosphatase (TRAP) (Santa Cruz Biotechnology, Santa Cruz, CA, USA) overnight at 4˚C. The sections were counterstained with hematoxylin and photographed using an Olympus photomicroscope (Tokyo, Japan). Mouse joint tissue was fixed in 4% paraformaldehyde, decalcified in ethylenediaminetetraacetic acid bone decalcifier, and embedded in paraffin. The resulting sections (7 μm) were stained with hematoxylin and eosin, Safranin O, and toluidine blue to detect proteoglycans. The number of positive cells was counted using Adobe Photoshop software (Adobe, USA) on high-power digital image (magnifcation: 400×). Positive cells were enumerated visually by three individuals, and the mean values were calculated.

## Collagen-specific IgG assay

Serum was collected 7 weeks after the first immunization for determination of the collagen-specific total IgG, IgG1, and IgG2a levels. CII (40 μg/mL) in coating buffer (0.05 M sodium carbonate anhydrous in distilled water, pH 9.6) was used to coat 96-well flat-bottom plates at 4˚C overnight. Serially diluted serum samples were incubated in the wells for 1 h at room temperature. The wells were washed with washing buffer (phosphate-buffered saline containing 50

mM Tris, 0.14 M NaCl, and 0.05% Tween 20), and a horseradish peroxidase (HRP)-conjugated goat anti-mouse IgG was added (Bethyl Laboratories, Montgomery, TX, USA). HRP activity was assayed by adding tetramethylbenzidine solution (eBioscience) and determining the absorbance at 450 nm.

## Enzyme-linked immunosorbent assay

The supernatant was collected 3 days after sulforaphane treatment. The IL-6, IL-17, and TNF-α levels were analyzed by sandwich enzyme-linked immunosorbent assay (ELISA) (R&D Systems). The absorbance at 405 nm was measured using an ELISA microplate reader (Molecular Devices).

## 3-(4,5-Dimethylthiazol-2-yl)-2,5-diphenyltetrasolium bromide assay

Cell viability was assessed by 3-(4,5-dimethylthiazol-2-yl)-2,5-diphenyltetrazolium bromide (MTT) assay based on the ability of mitochondria of viable cells to convert soluble MTT into an insoluble purple formazan reaction product. Cells were treated with MTT solution (5 mg/mL in Dulbecco's modified Eagle's medium without phenol red; Sigma-Aldrich) for 2 h. The MTT solution was aspirated and replaced with 200 mL/well dimethyl sulfoxide (DMSO). The absorbance at 540 nm of 100 mL of the reaction mixture was read at 540 nm. The results of two independent experiments performed in duplicate were pooled.

## Isolation and stimulation of splenocytes

Splenocytes were prepared from the spleens of normal C57BL/6 mice. Splenocytes were maintained in RPMI1640 medium supplemented with 5% fetal bovine serum (FBS; Gibco, Grand Island, NY, USA) before being stimulated with lipopolysaccharide (LPS; 100 ng/mL, Sigma-Aldrich) for 3 days and subjected to flow cytometry.

## Flow cytometry

Splenocytes were immunostained with various combinations of the following fluorescence-conjugated antibodies: APC-B220 (eBioscience, San Diego, CA, USA), PE-CD138, and FITC-GL-7 (BD Bioscience, San Diego, CA, USA). Events were collected on a FACSCalibur instrument (BD Bioscience). Data were analyzed using Flow Jo software (ver. 7.6; Treestar, Ashland, OR, USA).

## Isolation of peripheral blood mononuclear cells and synovial fluid mononuclear cells

We prepared peripheral blood mononuclear cells (PBMCs) from heparinized blood and synovial fluid mononuclear cells (SFMCs) from heparinized synovial fluids by standard density gradient centrifugation over Ficoll–Paque (GE Healthcare Biosciences, Uppsala, Sweden). Cells were cultured in RPMI-1640 medium (Gibco BRL, Carlsbad, CA, USA) containing penicillin (100 U/mL), streptomycin (100 μg/mL), and 10% FBS (Gibco BRL) that had been inactivated by heating to 55°C for 30 min. Suspensions of both cell types were dispensed into 48-well plates (Nunc, Rosklide, Denmark). Our study was approved by the institutional review board of Bucheon St. Mary's Hospital and was performed in accordance with the Helsinki II Declaration. All patients were informed and gave their written consent.

## Statistical analyses

Statistical analyses were conducted using the nonparametric Mann–Whitney *U*-test for comparisons of two groups and one-way analysis of variance with Bonferroni's *post hoc* test for comparisons of multiple groups. GraphPad Prism (ver. 5.01; GraphPad Software Inc., San Diego, CA, USA) was used for all analyses. $P < 0.05$ was the threshold for statistical significance. The data are presented as means ± standard deviation (SD).

## Results

### Sulforaphane attenuated RA

First, we investigated the effect of sulforaphane on inflammation and joint destruction in CIA mice, which is dependent on autoantibody production [16]. Sulforaphane treatment of CIA mice ameliorated arthritis compared with vehicle-treated mice (Figs 1A and S1). Histological

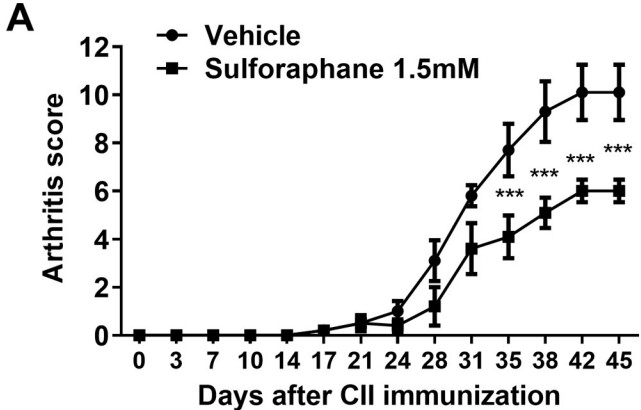

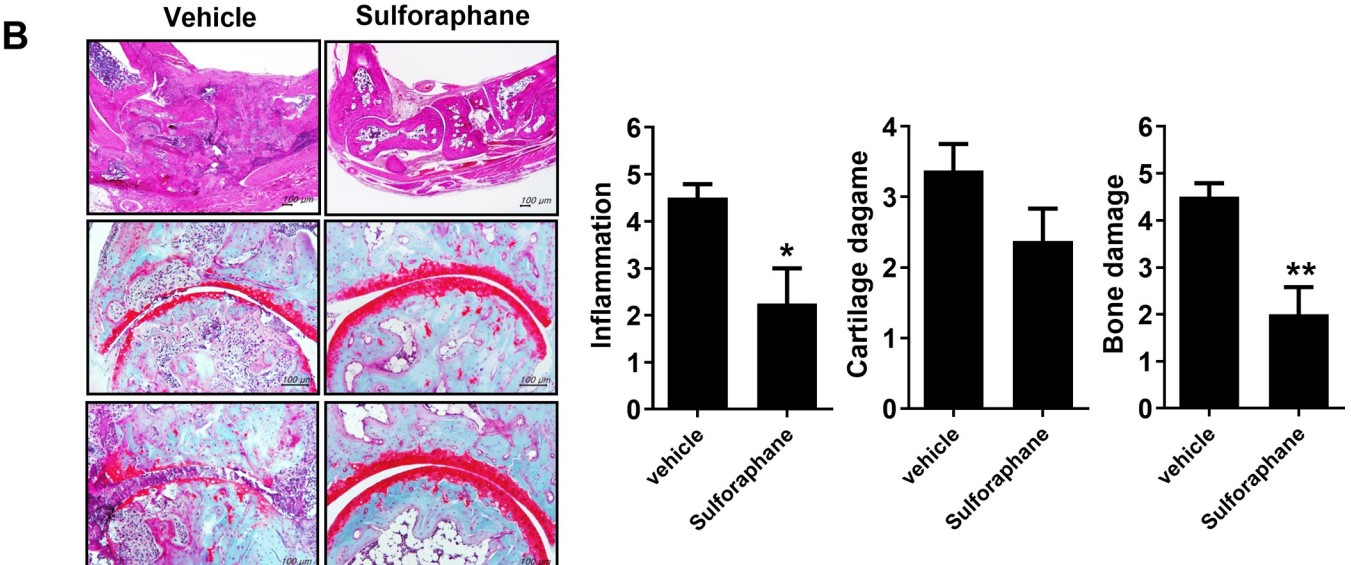

**Fig 1. Sulforaphane ameliorates collagen-induced arthritis (CIA) in mice.** (A) Arthritis score. Three weeks after CIA induction, CIA mice were intraperitoneally injected with sulforaphane three times per week (n = 5 per group); *$P < 0.001$. (B) Histological examination of the joints of CIA mice treated with sulforaphane. Mice were sacrificed on day 46 after CIA induction and tissue sections from the joints were stained with hematoxylin and eosin, and Safranin O. The histological scores of inflammations, cartilage damage, and bone damage are shown. ***$P < 0.001$, **$P < 0.01$.

sections of hind-paw joints showed that sulforaphane attenuated inflammation, cartilage damage, and bone erosion (Fig 1B).

### Anti-inflammatory effect of sulforaphane is associated with reduced expression of inflammatory cytokines in the joints of CIA mice

Next, we investigated the expression of inflammatory cytokines in vehicle- and sulforaphane-treated CIA mice. IL-6, IL-17, and TNF-α are proinflammatory cytokines that participate in the inflammatory process in the RA synovium and have systemic effects [17, 18]. Compared with vehicle-treated CIA mice, the joints from sulforaphane-treated CIA mice showed significantly fewer IL-6-, IL-17-, and TNF-α-expressing cells (Fig 2A). The receptor activator of NF-κB (RANK)/RANKL pathway plays a critical role in mediating articular bone erosion in RA. Focal bone loss in RA joints is mediated by osteoclasts, which express TRAP. Thus, we investigated the effect of sulforaphane on RANKL and TRAP expression in the joints of CIA mice.

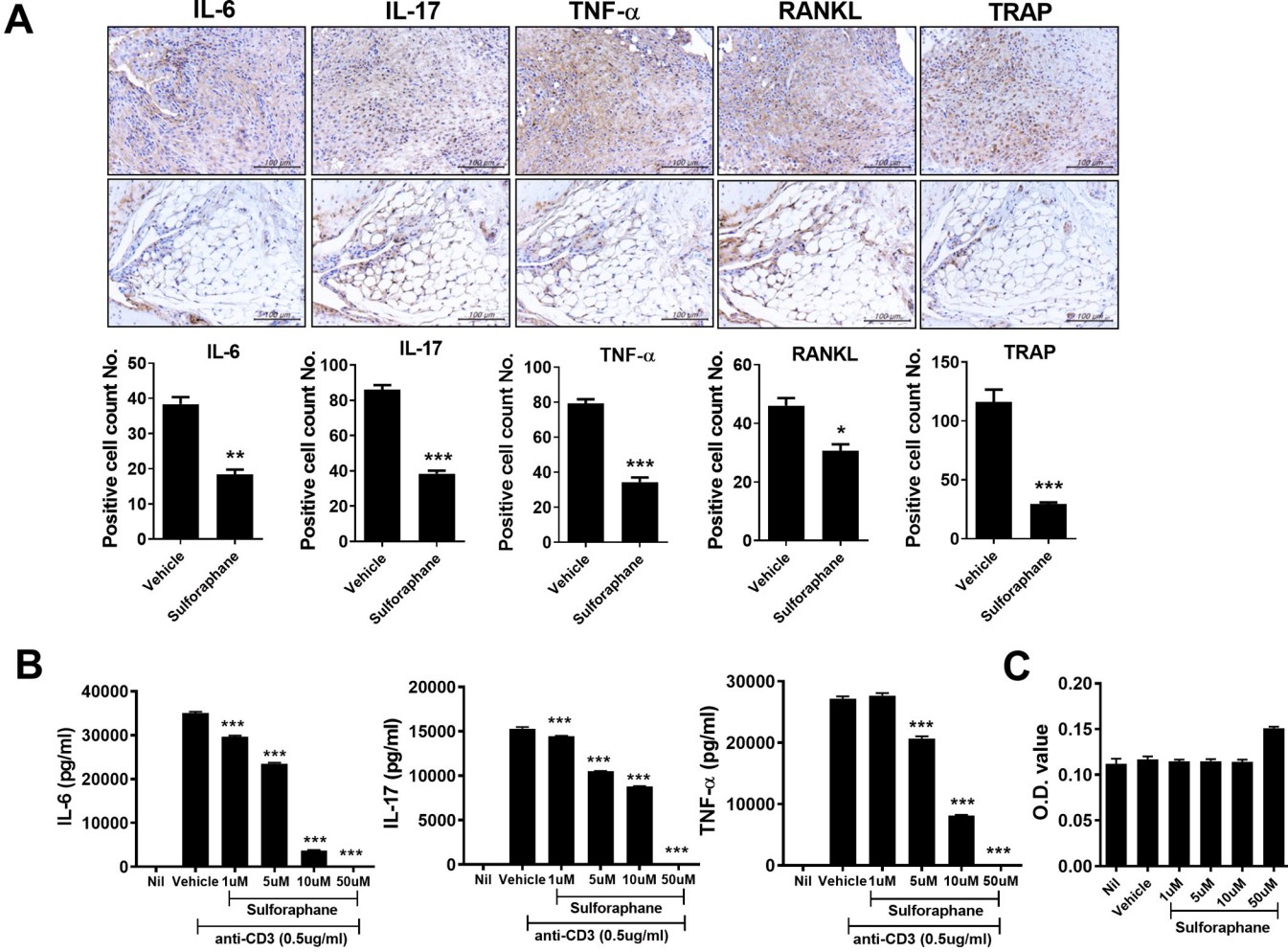

**Fig 2. Sulforaphane suppresses the expression of inflammatory cytokines in the joints of CIA mice.** (A) Immunohistochemical staining for interleukin (IL)-6, IL-17, tumor necrosis factor (TNF)-α, receptor activator of NF-κB ligand (RANKL), and tartrate-resistant acid phosphatase (TRAP) in the synovium of CIA mice. Data are means ± standard deviation (SD) of three independent experiments. (B) Splenocytes from C57BL/6 mice were cultured with sulforaphane (1, 5, 10, or 50 μM) for 72 h, and the IL-6, IL-17, and TNF-α levels in the culture supernatant were measured by ELISA. (C) Cell viability by MTT analysis. Data are means ± SD. ***$P < 0.001$, **$P < 0.01$, *$P < 0.05$ (bars represent means).

The reduced numbers of RANKL- and TRAP-positive cells in the joints of CIA mice treated with sulforaphane indicated suppression of osteoclastogenic activity (Fig 2A).

Next, we investigated the effect of sulforaphane on the production of proinflammatory cytokines. Splenocytes were isolated from C57BL/6 mice and cultured in the presence of an anti-CD3 antibody with or without sulforaphane (1 to 50 μM) for 72 h. The IL-6, IL-17, and TNF-α levels in the culture supernatant were decreased by sulforaphane in a dose-dependent manner (Fig 2B). By MTT assay, up to 50 μM sulforaphane was not cytotoxic to murine splenocytes (Fig 2C).

## Effect of sulforaphane on B-cell differentiation in vitro and Ig production

Sulforaphane suppressed differentiation into CD138+B220low plasma cells and GL7+B220 + germinal-center B cells compared with vehicle (Fig 3A). The differentiation of B cells into antibody-secreting plasma cells is an antigen-driven and cytokine-dependent process. This suppression of differentiation into plasma cells by sulforaphane was due to decreased production of proinflammatory cytokines (Fig 2B). Therefore, we investigated the effect of sulforaphane on autoantibody production *in vivo*; sulforaphane-treated CIA mice had lower circulating levels of CII-specific IgG, IgG1, and IgG2a (Fig 3B).

## Sulforaphane attenuated the production of IL-6, TNF-α, IL-17, and IgG in human PBMCs

Next, we confirmed that the anti-inflammatory and B cell-inhibitory effects of sulforaphane occur in human cells. Human PBMCs were cultured in the presence of absence of sulforaphane (dose, 1 to 10 μM) for 72 h in the presence of anti-CD3. Sulforaphane significantly inhibited the production of IL-6, TNF-α, and IL-17 by human PBMCs in a dose-dependent manner (Fig 4A). The production of IgG by RA SFMCs was inhibited by sulforaphane in a dose-dependent manner (Fig 4B). Therefore, sulforaphane attenuated RA by inhibiting the production of pathologic Igs.

## Discussion

In this study, sulforaphane reduced the clinical and histologic scores of CIA mice. The anti-arthritic and anti-inflammatory effects of sulforaphane were due to suppression of the differentiation of naïve cells into plasma cells and GC B cells. This is the first report that sulforaphane exerts an anti-arthritic effect by regulating B-cell differentiation. The efficacy of sulforaphane in inflammatory arthritis including RA and gout has been shown in several *in vitro* and *in vivo* studies [19–21]. However, these studies did not postulate the mechanisms underlying the anti-arthritic effect [20, 22, 23]. Several studies that have investigated the efficacy of sulforaphane on T cells have been published in the past. Kong JS et al previously demonstrated the anti-inflammatory property of sulforaphane in RA T cells, regarding T cell proliferation as well as inflammatory cytokines production such as IL-17 and TNFα [20]. Liang J et al recently reported T cell-specific effects of sulforaphane especially regarding Th17 cell differentiation (including the expression of transcriptional factor RORγt and their related cytokines such as IL-17 and IL-22) [24]. To our knowledge, no study has investigated the effect of sulforaphane on B-cell differentiation. Because plasma cells are not affected by conventional immunosuppressive drugs such as steroids, cyclophosphamide, and B-cell-depleting agents, our finding that sulforaphane suppresses their differentiation into plasma cells is encouraging and suggests that plasma cell-targeted treatment strategies for RA may be effective.

The CIA mice exhibited T cell- as well as B cell-mediated immune responses against CII. CII-specific antibodies are important pathogenic factors in CIA mice, because they induce the

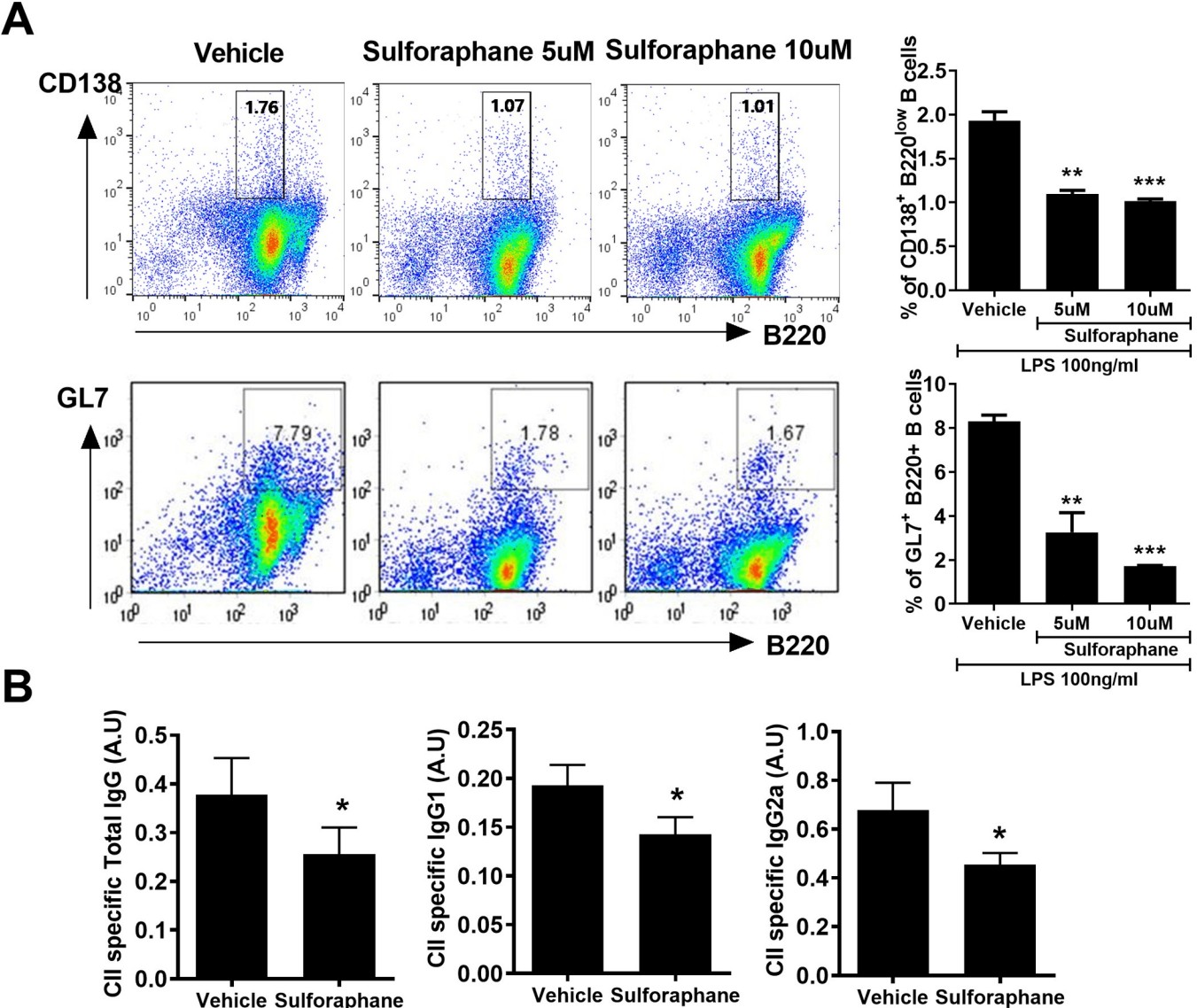

**Fig 3. Sulforaphane alters B cell differentiation.** (A) Splenocytes isolated from mice were analyzed by flow cytometry. The cells were stained with antibodies against CD138+B220low B cells for plasma B cells and GL7+B220+ B cells for germinal-center B cells. (B) Splenocytes were cultured with sulforaphane (5 or 10 μM) in the presence of 100 ng/mL lipopolysaccharide (LPS) for 3 days. The population of plasma B cells and germinal-center B cells were analyzed by flow cytometry. (B) Serum level of type II collagen (CII)-specific IgG 45 days after the first immunization. $^{***}P < 0.001$, $^{**}P < 0.01$, $^{*}P < 0.05$.

destruction of cartilage and activation of synovial cells [25, 26]. The presence of GCs indicates a pathologic RA synovium. GCs generate plasma cells as well as memory B cells. Thus, a GC-targeting strategy may be effective against RA. Here, we showed that sulforaphane treatment of murine splenocytes suppressed their differentiation into GL7+B220+ GC B cells as well as CD138+ plasma cells. Similarly, Dahada *et al.* reported that GC B cells play a pivotal role in CIA [27].

In addition to autoantibody production, B cells, like T cells and macrophages, produce proinflammatory cytokines [28, 29]. B cells induce immune-cell trafficking by producing chemokines, to which they respond via cell-surface chemokine receptors. Furthermore, B cells act as antigen-presenting cells during the development of T cell-mediated autoimmune diseases

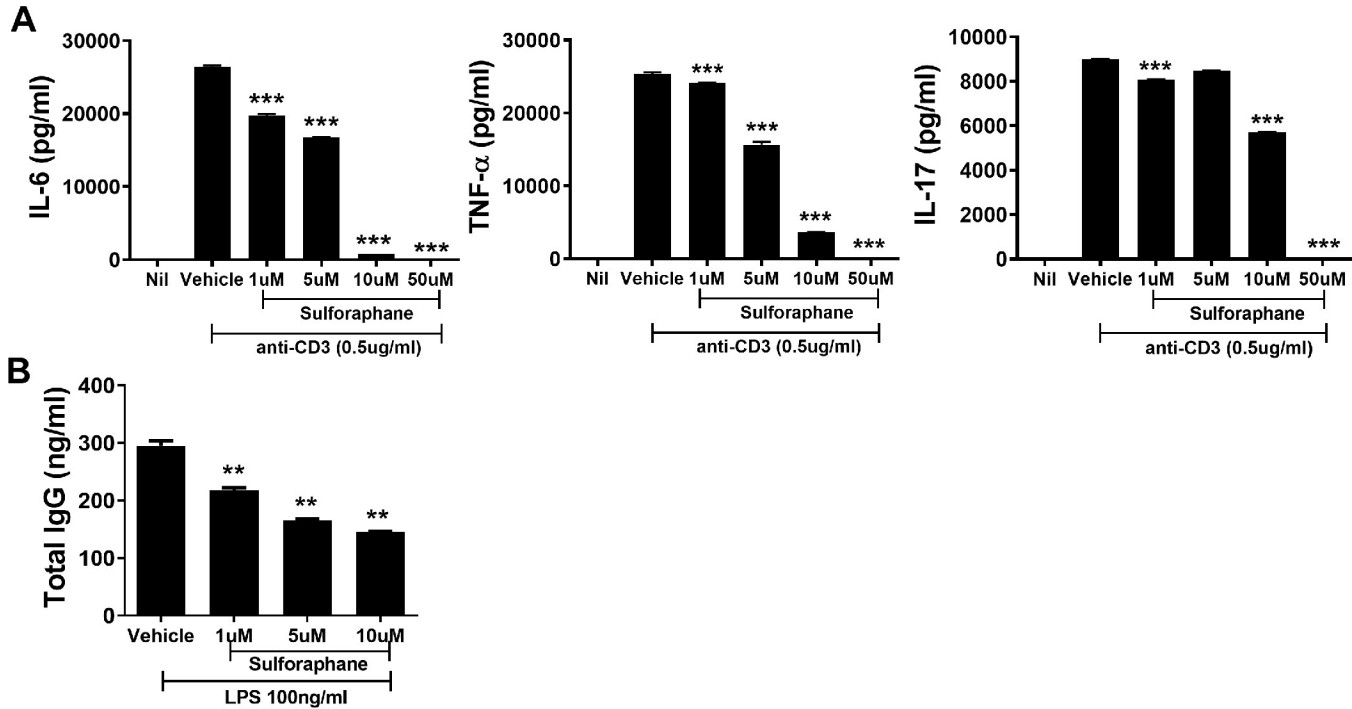

**Fig 4. Sulforaphane decreases IL-6, TNF-α, IL-17, and total IgG levels.** (A) Peripheral blood mononuclear cells from healthy controls were cultured in medium without or with sulforaphane (1, 5, 10 μM) for 72 h in the presence of 0.5 μg/mL anti-CD3. The levels of IL-6, TNF-α, and IL-17 in the culture supernatant were assayed. (B) Synovial fluid mononuclear cells from RA patients were cultured with sulforaphane (1, 5, 10 μM) for 72 h in the presence of 100 ng/mL LPS and the level of total IgG in the supernatant was determined. ***$P < 0.001$, **$P < 0.01$.

[30]. Classically, B-cell activation is dependent on T cells. Affinity maturation takes place in germinal centers after B cells encounter antigen and receive T-cell help via the CD40/CD154 interaction. However, B cells can also be activated in the absence of direct T-cell help via FcγRIIb and a Toll-like receptor [31, 32].

Our findings suggest suppression of B cells, irrespective of T-cell activity suppression, to be a rational anti-inflammatory strategy for RA. However, few B cell-directed therapies for RA are available; one example is rituximab, a CD20 monoclonal antibody. However, rituximab is typically used as a second-line therapy rather than as a primary therapeutic agent. Thus, a new B cell-targeting therapeutic agent is needed.

In conclusion, systemic administration of sulforaphane attenuated arthritis and histological inflammation in a murine model of RA. Sulforaphane significantly reduced the expression of proinflammatory cytokines in inflamed joints. This anti-inflammatory effect of sulforaphane was due to inhibition of B-cell differentiation into CD138+B22low plasma cells and GL7+B220 + GC B cells. This is the first report that the anti-inflammatory effect of sulforaphane is mediated by inhibition of the differentiation of B cells into GC B cells and plasma cells, which are implicated in the pathogenesis of RA.

## Supporting information

**S1 Fig. Sulforaphane-mediated inhibition of the development of CIA (A,B).** A,B. Reduction in arthritis score in CIA mice treated with Sulforaphane. CIA mice were injected intraperitoneal injected with sulforaphane (1.5mM, 12.8mg/mL/kg) every other day 7 weeks. (DOCX)

## Author Contributions

**Conceptualization:** Mi-La Cho.

**Data curation:** Su-Jin Moon, Jooyeon Jhun, Ji ye Kwon.

**Formal analysis:** Jaeyoon Ryu, Ji ye Kwon.

**Investigation:** Jooyeon Jhun, Mi-La Cho, Jun-Ki Min.

**Methodology:** Jooyeon Jhun, Jaeyoon Ryu, Se-Young Kim.

**Project administration:** Su-Jin Moon, Jooyeon Jhun, Mi-La Cho, Jun-Ki Min.

**Resources:** Mi-La Cho.

**Supervision:** Mi-La Cho, Jun-Ki Min.

**Validation:** Jaeyoon Ryu, KyoungAh Jung.

**Visualization:** Se-Young Kim, KyoungAh Jung.

**Writing – original draft:** Su-Jin Moon, Jooyeon Jhun.

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
