## [Decision Letter · Decision Letter 0]

10 Nov 2020

PONE-D-20-25189

The anti-arthritis effect of sulforaphane, an activator of Nrf2, is associated with inhibition of B cell differentiation.

PLOS ONE

Dear Dr. Cho,

Thank you for submitting your manuscript to PLOS ONE. After careful consideration, we feel that it has merit but does not fully meet PLOS ONE’s publication criteria as it currently stands. Therefore, we invite you to submit a revised version of the manuscript that addresses the points raised during the review process.

Both reviewers have issues with the details concerning the numbers of mice used in each experiment and reviewer 2 wants specific information about how mice were handled.  How do the second and third arthritis induction experiments compare to that listed in 1A?

Both reviewers share concerns about the specificity of suppressive effects on B cells, suggesting that the effects may either be broad (covering multiple cell types) or worse, non-specific in nature.  These points will need to be addressed before this manuscript can be considered for publication. 

Reviewer 1's comment that "There is really no experiments in the manuscript documenting that sulforaphane is actually an activator of Nrf2" could be handled either through experimentation or simply through highlighting evidence from the literature, but it should be acknowledged somehow.

I would point out that the flow cytometry data needs attention as well.  It is difficult to determine exactly how the regions outlining specific populations were chosen. I have attached a copy of figure 3 on which annotations have been made to point out specifics.

The first issue is that of gating.  How are the dot plots shown in Figure 3A gated?  The only antibodies listed were those for B220, CD138 and GL-7.  Sometimes it is helpful to include a plot of ex-vivo splenocytes as a comparison to demonstrate that your antibodies have identified the populations that you intended.  

The box used to define B220low/CD138+ plasma cells does not really seem to specify any definable population even in the vehicle group.  It appears to cover B220 negative cells.   If one wants to define B220lo and B220hi populations, then arguably the best division may be the orange line I have indicated on the annotation, which would then make the plasma cells fall into the magenta colored rectangle.  This might change the conclusions.

At any rate, a more complete description of the strategies used to define specific populations should be provided.

--

We look forward to receiving your revised manuscript.

Kind regards,

David Douglass Brand

Academic Editor

PLOS ONE

Journal Requirements:

2. PLOS ONE requires experimental methods to be described in enough detail to allow suitably skilled investigators to fully replicate and evaluate your study. See https://journals.plos.org/plosone/s/submission-guidelines#loc-materials-and-methods for more information.

To comply with PLOS ONE submission guidelines, in your Methods section, please provide a more detailed description of your immunohistochemistry, ELISA, and flow cytometry methodology. In addition, please ensure that you describe the sources, catalog numbers, and dilutions of all primary and secondary antibodies used in your study.

'The authors declare that there is no conflict of interest regarding the publication of this article.' 

We note that one or more of the authors are employed by a commercial company: "Impact Biotech".

Additional Editor Comments (if provided):

I would personally like to apologize for the length of time it took to get this review completed. Due to current circumstances, finding appropriately qualified reviewers for this work was particularly challenging but those agreeing to do so are indeed highly qualified for this subject area. Please look over their responses carefully as you decide how to proceed with this important work.

Reviewers' comments:

Reviewer's Responses to Questions

**Comments to the Author**

1. Is the manuscript technically sound, and do the data support the conclusions?

Reviewer #1: Yes

Reviewer #2: Partly

2. Has the statistical analysis been performed appropriately and rigorously? 

Reviewer #1: I Don't Know

Reviewer #2: Yes

3. Have the authors made all data underlying the findings in their manuscript fully available?

Reviewer #1: Yes

Reviewer #2: No

4. Is the manuscript presented in an intelligible fashion and written in standard English?

Reviewer #1: Yes

Reviewer #2: Yes

5. Review Comments to the Author

Reviewer #1: The anti-arthritis effect of sulforaphane, an activator of Nrf2, is associated with inhibition of B cell differentiation

The manuscript investigates the effect of sulforaphane on collagen-induced arthritis. Mice are treated with sulforaphane intraperitoneally and the investigators demonstrate that there is a decrease in severity of arthritis. This is associated with decreases in histologic inflammation, decreases in cytokines IL-6, IL-17, TNF-a, and TRAP. The investigators also demonstrate that human PBMs can have inhibition in production of IL-6, TNF-a, and IL-17 when cultures with sulforaphane. This is an important finding and may have therapeutic implications.

However, there are some problems with the manuscript.

The title is misleading. The effects of sulforaphane appear to be broad and cause inhibition of both T cell cytokines, as well as antibody production and inhibition of differentiation into plasma cells. Therefore, the focus on B cells is misleading.

There is really no experiments in the manuscript documenting that sulforaphane is actually an activator of Nrf2.

There was no clear indication as to how many mice were used in the experiments described in figure 1A. It is important to know the number of mice in order to evaluate the validity of the results.

For the flow experiments, it is also important to know how many mice were evaluated and what are the means of the populations evaluated.

Reviewer #2: This paper shows that sulforophan, known as an inhibitor of Nrf2, suppress collagen induced arthritis. It is claimed it do so based on its effects on B cells. Major points:

1) All data are not shown. Its not acceptable to base a study on an arthritis experiment with n=5. Apparently the experiment has ben run three times so this is a selected experiment. If the same experiment has been done 3 times it should be pooled and calculated together. The pooled data can be shown in the paper and the single experiments int he suppl information.

2) It should be clearly stated that the experiment was done blindly and distributed in the cages randomly, especially as it is well known there is a strong cage effect in DBA/1 mice. It should also be stated that all animal experiments follow the ARRIVE guideline. Of course only if this was the case.

3) The treatment has a profound effect on the inflammatory response. It is likely to ha ve very broad effects and it will be difficult to say exactly what is the specific effect. Basically all readouts a re secondary effects to something that this high dose of sulforophan is dong, whatever that is. Thus, it is not possible to claim that the effect on arthritis is due to effects on B cells as there is no evidence for this. The treatment is given 3 weeks after priming which means that the B cells have been primed and a full antibody response been developed. What will happen is that if these mice, due to this unknown "toxic" effect of the treatment does not develop arthritis it will secondarily, dur to less exposure of inflamed cartilage as well as a less powerful immune system give lower antibody titres. It can be predicted whatever is given to a mice leading to such a suppression of arthritis development.

In conclusion. If the arthritis data holds its a valuable report. But the authors need to make it very clear that they cannot say anything about the specific effects about sulforophan action as all evidenced data are secondary to the arthritis effect per se. Regarding human cells it seems to have profound inhibitory effect on cytokine production and I am afraid that these cells are not very happy.

6. PLOS authors have the option to publish the peer review history of their article (what does this mean?). If published, this will include your full peer review and any attached files.

Reviewer #1: No

Reviewer #2: No

---

## [Author Response · Author response to Decision Letter 0]

1 Dec 2020

Revised submission of manuscript PONE-D-20-25189

Dear David Douglass Brand

Editor-in-PLOS ONE

We thank the reviewers for their constructive and helpful comments concerning the manuscript. We have addressed the reviewers' concerns by either performing additional statistical analysis and revising the manuscript or explaining respectfully our rebuttal. The point-by-point replies are given in this letter. We hope that we have addressed satisfactorily all concerns raised by the reviewers, and that this manuscript is now suitable for publication.

Thank you again for the comments.

Sincerely yours,

Mi-La Cho, PhD

Department of Medical Lifescience, College of Medicine, The Catholic University of Korea, 222, Banpo-daero, Seocho-gu, Seoul, 06591, Republic of Korea

(Tel: 82 2 2258 7467, Fax: 82 2 2258 7473, E-mail address: iammila@catholic.ac.kr)

Comments from the editors and reviewers:

Reviewer #1: The anti-arthritis effect of sulforaphane, an activator of Nrf2, is associated with inhibition of B cell differentiation

The manuscript investigates the effect of sulforaphane on collagen-induced arthritis. Mice are treated with sulforaphane intraperitoneally and the investigators demonstrate that there is a decrease in severity of arthritis. This is associated with decreases in histologic inflammation, decreases in cytokines IL-6, IL-17, TNF-a, and TRAP. The investigators also demonstrate that human PBMs can have inhibition in production of IL-6, TNF-a, and IL-17 when cultures with sulforaphane. This is an important finding and may have therapeutic implications.

However, there are some problems with the manuscript.

The title is misleading. The effects of sulforaphane appear to be broad and cause inhibition of both T cell cytokines, as well as antibody production and inhibition of differentiation into plasma cells. Therefore, the focus on B cells is misleading.

Answer: Thanks for valuable comments. The authors agree with your comments and revised as you required. The title was revised as follows. “The anti-arthritis effect of sulforaphane, an activator of Nrf2, is associated with inhibition of B cell differentiation and inflammation cytokine.

There are really no experiments in the manuscript documenting that sulforaphane is actually an activator of Nrf2. 

Answer: Thanks for your comments. We have no experimtents in the manuscript. But, many

studies have shown that Sulforaphane is a potent activator of the endogenous anti-oxidant 

transcription factor nuclear factor (erythroid-derived 2)-like 2 (Nrf2). 

Reference

1. Eri Kubo, Bhavana Chhunchha, Prerna Singh, Hiroshi Sasaki & Dhirendra P. Singh. Sulforaphane reactivates cellular antioxidant defense by inducing Nrf2/ARE/Prdx6 activity during aging and oxidative stress. Scientific Reports, 7:14130 (2017)

2. Albena T. Dinkova-Kostova, Jed W. Fahey and Thomas W. Kensler. KEAP1 and done? Targeting the NRF2 pathway with sulforaphane. Trends in Food Science & Technology, 69:257-269(2017) 

3. Christine A. Houghton, Robert G. Fassett, and Jeff S. Coombes. Sulforaphane and Other Nutrigenomic Nrf2 Activators: Can the Clinician’s Expectation Be Matched by the Reality? Oxidative Medicine and Cellular Longevity, 7857186 (2016)

There was no clear indication as to how many mice were used in the experiments described in figure 1A. It is important to know the number of mice in order to evaluate the validity of the results. For the flow experiments, it is also important to know how many mice were evaluated and what are the means of the populations evaluated.

Answer: The authors agree with your comments and revised as you required. We added the method section. (marked by red color, Page_5_, line 16_). We analyzed the experiments 3 times and showed the most representative data. 

Reviewer #2: This paper shows that sulforophan, known as an inhibitor of Nrf2, suppress collagen induced arthritis. It is claimed it do so based on its effects on B cells. 

Major points:

1) All data are not shown. Its not acceptable to base a study on an arthritis experiment with n=5. Apparently the experiment has ben run three times so this is a selected experiment. If the same experiment has been done 3 times it should be pooled and calculated together. The pooled data can be shown in the paper and the single experiments in the suppl information.

Answer: Thanks for valuable comment which can clarify the method of present study. The experiment was performed three times. We analyzed the experiments 3 times and showed the most representative data. As the trend of the experiment was consistent, it was shown as a representative picture. 

2) It should be clearly stated that the experiment was done blindly and distributed in the cages randomly, especially as it is well known there is a strong cage effect in DBA/1 mice. It should also be stated that all animal experiments follow the ARRIVE guideline. Of course only if this was the case.

Answer: Thanks for comments. We performed grouping prior to Sulforaphane administration. To proceed with group separation, arthritis score and serum CII specific IgG2a measurement were analyzed. Then, group divide was conducted by ranking the results of two analyzes of arthritis score and CII specific IgG2a.

All experimental procedures were evaluated and conducted in accordance with the protocols approved by the Animal Research Ethics Committee at the Catholic University of Korea (Permit Number: CUMC 2016-0086-01). All procedures performed in this study followed the ethical guidelines for animal use.

3) The treatment has a profound effect on the inflammatory response. It is likely to have very broad effects and it will be difficult to say exactly what is the specific effect. Basically all readouts are secondary effects to something that this high dose of sulforophan is dong, whatever that is. Thus, it is not possible to claim that the effect on arthritis is due to effects on B cells as there is no evidence for this. The treatment is given 3 weeks after priming which means that the B cells have been primed and a full antibody response been developed. What will happen is that if these mice, due to this unknown "toxic" effect of the treatment does not develop arthritis it will secondarily, dur to less exposure of inflamed cartilage as well as a less powerful immune system give lower antibody titres. It can be predicted whatever is given to a mice leading to such a suppression of arthritis development.

In conclusion. If the arthritis data holds its a valuable report. But the authors need to make it very clear that they cannot say anything about the specific effects about sulforophan action as all evidenced data are secondary to the arthritis effect perse. Regarding human cells it seems to have profound inhibitory effect on cytokine production and I am afraid that these cells are not very happy.

Answer: Thanks for comments. We agree with the referee’s concern. We also had a lot of thinking. Because high ROS concentrations may lead, through oxidative damage of cellular constituents, to various disorders. It was thought that it was important to find a concentration that could sustain homeostasis without toxic effects. We could expand the research in future study. We will expand our research in future studies.

---

## [Editor Report · Decision Letter 1]

8 Dec 2020

PONE-D-20-25189R1

The anti-arthritis effect of sulforaphane, an activator of Nrf2, is associated with inhibition of B cell differentiation and inflammation cytokine.

PLOS ONE

Dear Dr. Cho,

Thank you for submitting your manuscript to PLOS ONE. After careful consideration, we feel that it has merit but does not fully meet PLOS ONE’s publication criteria as it currently stands. Therefore, we invite you to submit a revised version of the manuscript that addresses the points raised during the review process.

• Address the subject of the under-powered nature of the CIA study (both reviewers) either through the pooling of data or through new CIA experiments.  Your response did not properly address these previously.

• Provide the missing flow cytometric data in Figure three showing the nature of the splenocytes as taken from each group (sulforaphane treated and untreated) of CIA mice. 

We look forward to receiving your revised manuscript.

Kind regards,

David Douglass Brand

Academic Editor

PLOS ONE

Additional Editor Comments (if provided):

Both reviewers showed concerns regarding the numbers of mice used in the CIA experiment. The second reviewer suggested pooling data and supplying individual experiments as supplementary data. These concerns are shared by this editor in that under no circumstances will five mice per group properly power a CIA experiment. A minimum of twice that number is required, with even larger numbers necessary in experiments where the detection of subtle differences are required. However, your response to the reviewers did not address their concerns. I am afraid that providing more information than that or new data using larger numbers per group will be required before these answers can be met.

There is an additional problem in that it appears that some data are inadvertently missing from this publication. Figure 3 suggests that there are flow data available from cells analyzed ex vivo from sulforaphane treated mice. This is very confusing because the section is entitled "Effect of sulforaphane on B-cell differentiation *in vitro* and Ig production". The second and third sentences say "The results showed that the population of CD138+B220low plasma cells was significantly decreased by sulforaphane treatment in CIA mice, whereas the population of GL7+B220+ germinal-center B cells did not differ between the two groups (Fig. 3a). Then, in vitro experiments were performed to determine the effects of sulforaphane on B cell differentiation. " This suggests that the first set of data was a flow cytometric analysis of splenocytes from each group of mice demonstrating a reduction in CD138+B220low plasma cells but no reduction in GL7+B220+ germinal-center B cells. Figure 3a, which is referred to twice in this section appears to only show the in vitro data. The labeling demonstrates the concentrations (5 and 10 µM sulforaphane) and clearly shows in vitro data due to the LPS stimulated absence of T cells (very few double negative cells) under any condition. What happened to the flow data demonstrating a reduction in CD138+B220low plasma cells but no reduction in GL7+B220+ germinal-center B cells taken from CIA mouse spleens?

Other points are

The title that you suggested including the words "and inflammation cytokine" is not proper English usage. Perhaps "The anti-arthritis effect of sulforaphane, an activator of Nrf2, is associated with inhibition of both B cell differentiation and the production of inflammatory cytokines"

Strike the word "etiological" from the last sentence of the abstract or provide more information about what your were trying to say.

I am aware that this decision is a very difficult one to resolve, but since this editor agrees with the concerns of the two reviewers regarding properly powering the study, it cannot be ignored as you have done in your response.

---

## [Author Response · Author response to Decision Letter 1]

2 Jan 2021

Comments from the editors and reviewers:

Reviewer #1: The anti-arthritis effect of sulforaphane, an activator of Nrf2, is associated with inhibition of B cell differentiation

The manuscript investigates the effect of sulforaphane on collagen-induced arthritis. Mice are treated with sulforaphane intraperitoneally and the investigators demonstrate that there is a decrease in severity of arthritis. This is associated with decreases in histologic inflammation, decreases in cytokines IL-6, IL-17, TNF-a, and TRAP. The investigators also demonstrate that human PBMs can have inhibition in production of IL-6, TNF-a, and IL-17 when cultures with sulforaphane. This is an important finding and may have therapeutic implications.

Answer: We do appreciate the valuable comments from the reviewers about the manuscript. We have gone through the suggestive comments from all the reviewers and incorporated in the final revised version.

However, there are some problems with the manuscript.

The title is misleading. The effects of sulforaphane appear to be broad and cause inhibition of both T cell cytokines, as well as antibody production and inhibition of differentiation into plasma cells. Therefore, the focus on B cells is misleading.

Answer: Thanks for valuable comments. The authors agree with your comments and revised as you required. The title was revised as follows. “The anti-arthritis effect of sulforaphane, an activator of Nrf2, is associated with inhibition of both B cell differentiation and the production of inflammatory cytokines.”

There are really no experiments in the manuscript documenting that sulforaphane is actually an activator of Nrf2. 

Answer: Thanks for your comments. We have no experiments in the manuscript. But, many 

studies have shown that Sulforaphane is a potent activator of the endogenous anti-oxidant 

transcription factor nuclear factor (erythroid-derived 2)-like 2 (Nrf2). 

Reference

1. Eri Kubo, Bhavana Chhunchha, Prerna Singh, Hiroshi Sasaki & Dhirendra P. Singh. Sulforaphane reactivates cellular antioxidant defense by inducing Nrf2/ARE/Prdx6 activity during aging and oxidative stress. Scientific Reports, 7:14130 (2017)

2. Albena T. Dinkova-Kostova, Jed W. Fahey and Thomas W. Kensler. KEAP1 and done? Targeting the NRF2 pathway with sulforaphane. Trends in Food Science & Technology, 69:257-269(2017) 

3. Christine A. Houghton, Robert G. Fassett, and Jeff S. Coombes. Sulforaphane and Other Nutrigenomic Nrf2 Activators: Can the Clinician’s Expectation Be Matched by the Reality? Oxidative Medicine and Cellular Longevity, 7857186 (2016)

There was no clear indication as to how many mice were used in the experiments described in figure 1A. It is important to know the number of mice in order to evaluate the validity of the results. For the flow experiments, it is also important to know how many mice were evaluated and what are the means of the populations evaluated.

Answer: The authors agree with your comments and revised as you required. We added the method section. (marked by red color, Page_5_, line 17_). We analyzed the experiments 3 times and showed the most representative data. We provided all the arthritis score graphs. These graphs are shown below.

Reviewer #2: This paper shows that sulforophan, known as an inhibitor of Nrf2, suppress collagen induced arthritis. It is claimed it do so based on its effects on B cells. 

Answer: We really appreciate your time and effort to edit our manuscript. In this revised manuscript, we have resolved most of the issues raised by the reviewers as you can see in our response to their comments below.

Major points:

1) All data are not shown. Its not acceptable to base a study on an arthritis experiment with n=5. Apparently the experiment has ben run three times so this is a selected experiment. If the same experiment has been done 3 times it should be pooled and calculated together. The pooled data can be shown in the paper and the single experiments in the suppl information.

Answer: Thanks for valuable comment which can clarify the method of present study. The experiment was performed three times. We analyzed the experiments 3 times and showed the most representative data. As the trend of the experiment was consistent, it was shown as a representative picture. 

2) It should be clearly stated that the experiment was done blindly and distributed in the cages randomly, especially as it is well known there is a strong cage effect in DBA/1 mice. It should also be stated that all animal experiments follow the ARRIVE guideline. Of course only if this was the case.

Answer: Thanks for comments. We performed grouping prior to Sulforaphane administration. To proceed with group separation, arthritis score and serum CII specific IgG2a measurement were analyzed. Then, group divide was conducted by ranking the results of two analyzes of arthritis score and CII specific IgG2a.

All experimental procedures were evaluated and conducted in accordance with the protocols approved by the Animal Research Ethics Committee at the Catholic University of Korea (Permit Number: CUMC 2016-0086-01). All procedures performed in this study followed the ethical guidelines for animal use.

3) The treatment has a profound effect on the inflammatory response. It is likely to have very broad effects and it will be difficult to say exactly what is the specific effect. Basically all readouts are secondary effects to something that this high dose of sulforophan is dong, whatever that is. Thus, it is not possible to claim that the effect on arthritis is due to effects on B cells as there is no evidence for this. The treatment is given 3 weeks after priming which means that the B cells have been primed and a full antibody response been developed. What will happen is that if these mice, due to this unknown "toxic" effect of the treatment does not develop arthritis it will secondarily, dur to less exposure of inflamed cartilage as well as a less powerful immune system give lower antibody titres. It can be predicted whatever is given to a mice leading to such a suppression of arthritis development.

In conclusion. If the arthritis data holds its a valuable report. But the authors need to make it very clear that they cannot say anything about the specific effects about sulforophan action as all evidenced data are secondary to the arthritis effect perse. Regarding human cells it seems to have profound inhibitory effect on cytokine production and I am afraid that these cells are not very happy.

Answer: Thanks for comments. We agree with the referee’s concern. As pointed out by the reviewers, there is insufficient direct evidence for inducing the immunomodulatory effect of B cells in the animal model of rheumatoid arthritis as a direct response to the drug. Antioxidants such as sulforafan can appear when excessive inhibition of oxidative stress is accompanied by an inflammatory response. Therefore, when using antioxidant drugs, dosages that do not have cellular toxic effects and inflammatory responses should be used. However, there seems to be a difficulty because the dosage used for the drug is not proportional even between in vitro and in vivo experiments. As a reviewer's opinion, it seems difficult to identify directly in vivo that the therapeutic effect in an animal model is a direct regulatory response to B cells from our results. In the future, we will take this part into account and try to confirm it in subsequent studies. Thank you very much for your review comment.

---

## [Editor Report · Decision Letter 2]

12 Jan 2021

The anti-arthritis effect of sulforaphane, an activator of Nrf2, is associated with inhibition of both B cell differentiation and the production of inflammatory cytokines.

PONE-D-20-25189R2

Dear Dr. Cho,

We’re pleased to inform you that your manuscript has been judged scientifically suitable for publication and will be formally accepted for publication once it meets all outstanding technical requirements.

Kind regards,

David Douglass Brand

Academic Editor

PLOS ONE

Additional Editor Comments (optional):

So long as the two additional severity scatter plots/Arthritis score graphs which were provided in your second response (Revision 2) are provided to the readers in the supplemental data section, the manuscript is acceptable.

However,

The wording is still confusing on page 11 would suggest removing lines 2 through 5 entirely.

Instead, start the section with line 6 but remove the word "Then"
---

## [Editor Report · Acceptance letter]

3 Feb 2021

PONE-D-20-25189R2 

The anti-arthritis effect of sulforaphane, an activator of Nrf2, is associated with inhibition of both B cell differentiation and the production of inflammatory cytokines. 

Dear Dr. Cho:

I'm pleased to inform you that your manuscript has been deemed suitable for publication in PLOS ONE. Congratulations! Your manuscript is now with our production department. 

Kind regards, 

on behalf of

Dr. David Douglass Brand 

Academic Editor

PLOS ONE